# Treatment outcome of neonatal sepsis and associated factors among neonates admitted to neonatal intensive care unit in public hospitals, Addis Ababa, Ethiopia, 2021. Multi-center cross-sectional study

**Asalef Endazanaw**[1], **Tefera Mulugeta**[2☯], **Fikertemariam Abebe**[2☯], **Yohannes Godie**[3], **Yitayal Guadie**[3☯], **Dires Birhanu**[4☯], **Esmelealem Mihretu**[3☯] *

1 Yekatit 12 Hospital, Addis Ababa, Ethiopia, 2 College of Health Science, Addis Ababa University, Addis Ababa, Ethiopia, 3 Colleges of Medicine and Health Science, Debre Markos University, Debre Markos, Ethiopia, 4 College of Health Science, Dila University, Dilla, Ethiopia

☯ These authors contributed equally to this work.
* esmemahi2119@gmail.com

**Data Availability Statement:** All relevant data are within the paper.

## Abstract

### Background

Globally, neonatal sepsis is the leading cause of neonatal mortality and morbidity, particularly in developing countries. Despite studies that revealed the prevalence of neonatal sepsis in developing countries, the outcome of the diseases, barriers for poor outcomes were inconclusive. The aim of this study was to assess the treatment outcome of neonatal sepsis and its associated factors among neonates admitted to neonatal intensive care unit in public hospitals, Addis Ababa, Ethiopia, 2021.

### Methods

A cross-sectional study was carried out from February 15 to May 10, 2021 on 308 neonates admitted to neonatal intensive care units of Addis Ababa city public hospitals. Hospitals and study participants were selected by lottery and systematic random sampling techniques, respectively. Data were collected through face-to-face interviews with a structured, pre-tested questionnaire and by reviewing both the maternal and newborn profile cards. Epi-data version 4.6 was used to enter the collected data, which was then exported to SPSS version 26 for analysis. The 95% CI odds ratio is used to determine the direction and strength of the association between the dependent and independent variables.

### Results

Among the total study 308 neonates, 75(24.4%) were died. Regarding the poor treatment outcome of neonatal sepsis, neonates whose mothers <37 weeks of gestational age (AOR = 4.87, 95% CI: 1.23–19.22), Grunting (AOR 6.94: 1.48–32.54), Meconium amniotic stained (AOR = 3.03, 95% CI: 1.02–9.01), Duration of rupture of membrane >18hours (AOR = 3.66,

**Funding:** The author(s) received no specific funding for this work.

**Competing interests:** The authors have declared that no competing interests exist.

95% CI: (1.20–11.15), Hypertensive PIH/ Eclampsia (AOR = 3.54, 95% CI: 1.24–10.09), Meropenum (AOR = 4.16, 95% CI: 1.22–14.21) and CRP positive result (AOR = 5.87, 95% CI: 1.53–22.56) were significantly associated with poor treatment outcome of neonatal sepsis.

## Conclusion and recommendation

The treatment outcomes of neonates were 75.6% recovered and 24.4% died. In this setting, empirical treatment was the cornerstone for managing neonatal sepsis. Professionals who are working in labor and delivery ward screened for mothers preeclampsia and duration of rupture of membrane >18hrs /PROM/ treated with antihypertensive drug and antibiotics for the prevention of neonatal sepsis.

## Introduction

Neonatal sepsis is characterized as an infectious disease syndrome with a clinically suspected or culture-confirmed infection occurring within the first 28 days of life. It is a prevalent critical illness in the neonatal intensive care unit (NICU) [1]. One of the primary causes of morbidity and mortality in newborns is neonatal sepsis, a systemic infection that affects infants up to 28 days of age [2].

Around the world, newborn sepsis is thought to be the cause of 26% of fatalities in children under the age of five, with Sub-Saharan Africa (SSA) having the highest mortality rates. Neonatal mortality is unevenly distributed in Sub-Saharan Africa, where it is estimated that 49.6% of all under-five deaths occurred in 2013 [3]. Additionally, 30–50% of newborn mortality in underdeveloped nations and 15% of all neonatal deaths worldwide are attributable to neonatal sepsis. In Sub-Saharan Africa, neonatal sepsis is responsible for 17% of neonatal mortality. Neonatal sepsis, which causes over 37% of newborn mortality in Ethiopia and more than one-third of all neonatal deaths, is to blame [4].

One of the global initiatives of WHO is to reduce infant and under-five mortality in African countries to as low as 12/1000 and 25/1000, respectively, by the year 2030. The key to accomplishing this would be to improve the management and prevention of severe infections and preterm births [5].

In Ethiopia study shows that, small gestational age, being male, out born, not having breast-feed and lower Apgar score in the first 5 minute were identified as associated factors for the poor treatment outcome of preterm neonates admitted in NICU [6]. According to the 2019 Mini Ethiopian Demographic Health Survey (MEDHS) report, the neonatal mortality rate (NMR) is 30/1000 live births, a slight decrease from the 2011 EDHS report of 37/1000 live births. This high number of deaths is largely due to neonatal sepsis [5, 7]. In a number of developing countries, identification of factors for neonatal sepsis and treatment of neonates with sepsis is not satisfactory. Moreover, reports from low income countries revealed inconsistencies in the prevalence, risk factors, and mortality from that of developed countries. Identification of risk factors and timely initiation of treatments can significantly decrease neonatal mortality and morbidity [8]. The outcomes of neonatal sepsis treatment vary between hospitals with different setups. Early detection and treatment are required to save the lives of our children and grandchildren. This may necessitate the use of expertise to identify common risk factors, antimicrobial use patterns, and clinical outcome treatment of neonatal sepsis. Therefore

the purpose of this study was to assess treatment outcome and associated factors of neonatal sepsis among neonates admitted to neonatal intensive care unit in public hospitals, Addis Ababa, Ethiopia.

## Methods and materials

### Study design and setting

This was a facility-based quantitative cross-sectional study carried out from February 15 to May 10, 2021 across Addis Ababa city public hospitals. Addis Ababa is the capital city of Ethiopia and seat of African Union and the United Nations World Economic Commission for Africa. It covers an area of 527 square kilometers and has 11 sub cities [9]. According to a population projection value for 2020 the city has an estimated population of 4.8 million.

The city has 12 public Hospitals among these, 11 hospitals having neonatal intensive care unit. Among these 6 were under Addis Ababa Health Bureau, 5 were under ministry of health and 1 was under Addis Ababa University (Tikur Anbessa Specialized Hospital) [10]. The study was conducted in four Addis Ababa public Hospitals (36%) selected by lottery method. These selected hospitals are Gandhi Memorial hospital (GMH), St peter Specialized hospital (SPSH), Tikur Anbessa special hospital (TASH) and Yekatit 12 hospital medical college (Y12HMC).

### Sample size and sampling procedure

The sample size was determined using a single population–proportion formula by assuming a proportion (P) with culture-proven neonatal sepsis of 23.9%, the previous study in Ethiopia [11], a 5% margin of error (d), 10% nonresponse rate, n = $(Za/2)^2 p(1 - p)/(d)^2$ and the result was 308 neonates. Of the Eleven public hospitals in Addis Ababa city, four public hospitals (36%) were randomly selected for this study using a lottery method to.

Before the actual data collection started, the total number of neonates diagnosed as sepsis and admitted to NICU monthly was reviewed in each of the selected hospitals. Then the total sample size of the study was allocated proportionally to each selected hospital based on their previous number of neonates diagnosed as sepsis admitted to NICU. Neonates that fulfill eligibility criteria were recruited from respective study hospitals. Finally, 308 neonates diagnosed with sepsis were selected by using systematic random sampling technique from February 15, 2021 to May 10, 2021 with 'k' interval of 2. During the data collection, the first participant was selected by lottery method.

**Study population.** During the study period, all neonate patients admitted to NICUs of Addis Ababa selected public hospitals were diagnosed with neonatal sepsis.

### Inclusion criteria

The study included neonates with clinical diagnosis of sepsis based on the following two risk factors and/or clinical features of bacterial infections.

**Risk factors include** low birth weight (<2500 grams) or prematurity (<37 weeks of gestation age), febrile illness in the mother within 2 weeks prior to delivery, foul-smelling discharge and/or meconium stained amniotic liquid, prolonged rupture of membranes >18 hours, suspected chorioamnionitis, prolonged labor (> 24 hours), and perinatal asphyxia (Apgar score <4 at 1 minute).

**Clinical features of sepsis include** poor reflexes, lethargy, respiratory distress, bradycardia, apea, fever, convulsions, abdominal distension, and bleeding.

## Data collection instrument and procedures

Data were gathered using a structured questionnaire and in face to face interviews. The data collection tools were adapted from various sources of information [3, 10–13] and by reviewing both maternal and newborns profile cards. To ensure consistency, the questionnaire was created in English first, and then translated into Amharic and then back into English by language experts. The data was collected in day the day time shift from 8:00 a.m. to 5:30 p.m. after obtaining consent from each participant prior to data collection in the hospitals after finishing the services and returning home. Data was recruited by eight trained BSc nurses and they were supervised by four senior nurses having previous experience in data collection. Training was provided on data collection procedures; including how to conduct interviews, administer questionnaires, obtain consent, maintain confidentiality, and respect the rights of participants. Continuous follow up and supervision was made by principal investigator throughout the data collection period from February 15, 2021 to May 10, 202.

## Data quality control

Supervisors and data collectors were trained on how and what information they should collect from the targeted data sources to ensure data quality. Expertise was given a tool to check the content's validity and accuracy. It was pre-tested on 5% (n = 15) of similar mothers outside the study area at Zewditu Memorial Hospital to assess its completeness, clarity, length, skip patterns, and correctness of filled questioners. The questionnaire was modified based on the results of the pretest. Data was collected by trained health professionals from other units of the health facility.

## Data analysis procedure

Data were entered into EPI Data version 4.6 and analyzed in SPSS Software version 26. Bi-variable analysis was used to examine the relationship between each independent variable and the outcome variable. To account for all potential confounders, all variables with p-values $\leq 0.25$ were included in the multivariable model. The linear correlation among the independent variables was also examined using multi-co linearity. The degree of association between dependent and independent variables was determined using an odds ratio with a 95% confidence interval and a p-value $\leq 0.05$.

# Results

## Socio-demographic characteristics of respondents

All 308 required study participants were interviewed, with a 100% response rate. Among those who responded, 121 (39.3%) of the mothers were between the ages of 25 and 29. The mother's average age was 29.37(±5.16) years. Almost all of the respondents, 296 (96.2%), were married. Around 267 (63.6%) of mothers had secondary or higher education. The average monthly household income of the study participants was 7501 Ethiopian Birr (136 (44.2%) of government employees and 84 (27.3%) of respondents. The mean and standard deviation of household income were 5775.8 (±3543.2) years.

## Obstetric and neonatal health related factors of the study participants for their neonates

Almost all the respondents, 303 (98.4%), had an ANC follow-up. 221 (71.8%) of those who received ANC follow-up had four or more visits. The majority of mothers, 247 (80.2%),

**Table 1. Obstetric related factors of treatment outcome of neonatal sepsis among neonates admitted to neonatal intensive care unit in public hospitals, Addis Ababa, Ethiopia, 2021 (n = 308).**

| Variables | Frequency | Percentage (%) |
|---|---|---|
| **Parity** | | |
| Primi | 126 | 40.9 |
| Multi | 182 | 59.1 |
| **ANC visit** | | |
| Yes | 303 | 98.4 |
| No | 5 | 1.6 |
| **Number of ANC visit (n = 303)** | | |
| 1–3 | 82 | 26.6 |
| ≥4 | 221 | 71.8 |
| **Place of delivery** | | |
| Home | 5 | 1.6 |
| Health center | 56 | 18.2 |
| Hospital | 247 | 80.2 |
| **Maternal complications** | | |
| Yes | 260 | 84.4 |
| No | 48 | 15.6 |
| **PROM (n = 260)** | | |
| Yes | 170 | 55.2 |
| No | 90 | 29.2 |
| **Chorioamnionitis (n = 260)** | | |
| Yes | 42 | 16.2 |
| No | 218 | 83.8 |
| **Hypertensive (PIH, Eclampsia (n = 260)** | | |
| Yes | 75 | 24.4 |
| No | 185 | 60.1 |
| **Gestational Age** | | |
| < 37 week | 158 | 51.3 |
| ≥37 week | 150 | 48.7 |

delivered their newborns in a hospital. Half of the 157 mothers with their newborns were 37 weeks gestational age (Table 1).

In terms of birth weight, 163 (52.9%) of neonates weighed 2.5 kg. The majority of neonates, 262 (85.1%), were admitted when they were older than 30 minutes (Table 2).

## Treatment and laboratory findings of neonatal sepsis

Regarding the treatment outcomes, the majority of neonates 233 (75.6%) recovered from their condition with improvement, and 75 (24.4%) died (Table 3).

Almost all neonates were administered the combination of ampicillin and gentamicin as a first line (Fig 1).

In this study, the possible causes of neonatal deaths are 35(46.67%), cardio respiratory arrest secondary to respiratory problems and 14(18.67%) sepsis (Fig 2).

## Predictor variable of treatment outcome of neonatal sepsis

The association of the independent and dependent variable were first tested by using bi-variable analysis which (P≤0.25) were tested in the final multivariable analysis to see their

**Table 2. Neonatal related factors of treatment outcome of neonatal sepsis among neonates admitted to neonatal intensive care unit in public hospitals, Addis Ababa, Ethiopia, 2021 (n = 308).**

| Variables | Frequency | Percentage (%) |
|---|---|---|
| Birth weight | | |
| <2.5kg | 145 | 47.1 |
| ≥2.5kg | 163 | 52.9 |
| Sex | | |
| Male | 188 | 61.0 |
| Female | 120 | 39.0 |
| Age of the neonates admitted to NICU | | |
| <30 minutes | 46 | 14.9 |
| ≥30 minutes | 262 | 85.1 |
| Meconium stained | | |
| Yes | 70 | 22.7 |
| No | 238 | 77.3 |
| Grunting | | |
| Yes | 205 | 66.6 |
| No | 103 | 33.4 |
| Chest in drawing | | |
| Yes | 67 | 21.8 |
| No | 241 | 78.2 |
| Unable to feed (failure to suck) | | |
| Yes | 223 | 72.4 |
| No | 85 | 27.6 |
| Temperature >37.5 or <35.5oC | | |
| Yes | 171 | 55.5 |
| No | 137 | 45.5 |

significant association with their treatment outcome of neonatal sepsis. Accordingly, as shown in Table 4 below those bi-variable regression associated with the crude odds ratios (COR) treatment outcome of neonatal sepsis such as education status of the mothers, place of delivery, gestational age < 37weeks, birth weight of the neonate, age of the neonate admitted to NICU, sever chest in drawing, grunting, un able to feed, temperature, Chorioamnaties, meconium amniotic stained, PROM, Hypertensive PIH/ Eclampsia, meropenum, vancomycin, metronidazole, CBC and CRP result.

In Multivariable analysis results showed that, there was statistically significance association found between) Poor treatment outcome of neonatal sepsis parameters which showed p-value of below 0.05 were Preterm babies admitted to NICU (gestational age), grunting, meconium amniotic stained, duration of rupture of membrane >18hours (PROM), Hypertensive PIH/ Eclampsia, meropenum and CRP result.

## Discussion

This study aimed to assess treatment outcome of neonatal sepsis and its associated factors among newborns delivered in public hospitals in Addis Ababa, Ethiopia. In this study, the prevalence of poor neonatal sepsis treatment outcome was 24.4% with a 95% CI among newborns admitted to NICU in public hospitals in Addis Ababa (19.5–29.2). This finding is higher compared to a study conducted in Bahir dar (16%) [12]. This deference could be attributed to

**Table 3. Treatment and laboratory findings of neonatal sepsis among neonates admitted to neonatal intensive care unit in public hospitals, Addis Ababa, Ethiopia, 2021 (n = 308).**

| Variables | Frequency | Percentage (%) |
|---|---|---|
| **Treatment outcome** | | |
| Died | 75 | 24.4 |
| Recovered | 233 | 75.6 |
| **Did the neonate on oxygen** | | |
| Yes | 252 | 81.8 |
| No | 56 | 18.2 |
| **Did the neonate had started feeding** | | |
| Yes | 257 | 83.4 |
| No | 51 | 16.6 |
| **Breast milk** | | |
| Yes | 255 | 82.8 |
| No | 53 | 17.2 |
| **Formula milk** | | |
| Yes | 34 | 11.0 |
| No | 274 | 89.0 |
| **NGT feeding** | | |
| Yes | 157 | 51.0 |
| No | 151 | 49.0 |
| **Onset of sepsis** | | |
| Early | 291 | 94.5 |
| Late | 17 | 4.5 |
| **CBC result** | | |
| Normal | 113 | 36.7 |
| Drop/raise | 195 | 63.3 |
| **CRP result** | | |
| Non-reactive | 70 | 23.5 |
| Reactive | 228 | 76.5 |
| **CSF result** | | |
| Normal | 27 | 8.8 |
| Drop/raise | 281 | 91.2 |
| **Blood culture** | | |
| Normal | 52 | 16.9 |
| Drop/raise | 256 | 83.1 |
| **X-ray** | | |
| Normal | 28 | 9.1 |
| Abnormal finding | 280 | 90.9 |

the way neonatal sepsis has been used to confirmatory blood culture to assert neonatal sepsis to the study that is done in some facility. In addition to this, NICU services at Felegehowot referral hospital are organized with personnel and equipment based on newborn conditions (severity) and classified as level-1(basic), level-2 (specialty) and level-3 (subspecialty) and study design (retrospective follow up that all deaths might not be documented. Many of the complications of preterm birth that result in poor in-hospital outcomes are related to gestational age, with premature infants being more vulnerable. This finding is nearly identical to a study conducted in Debrezeyt, Ethiopia, where 26% [13]. This finding is lower compared to a

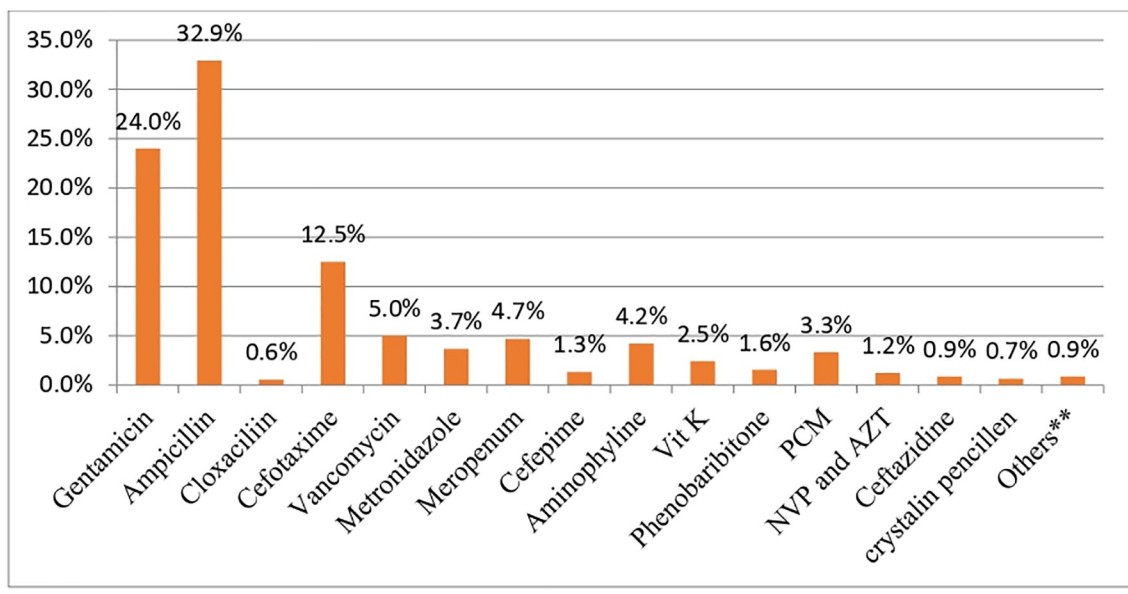

**Fig 1. Antibiotics administration among neonates admitted to neonatal intensive care unit in public hospitals, Addis Ababa, Ethiopia, 2021. Others**\*\*: fluconazole, esomeprazole, TTC, phototherapy.

study conducted in Nigeria were (34%) [14]. This could be related with use of advanced confirmatory blood culture to establish neonatal sepsis in the Nigeria, which is difficult to apply in this study area due to the lack of some facilities. Another possible explanation for this variation is the variation in health facility and sample size across studies.

Neonatal sepsis treatment outcomes were 5 times more likely to be bad in newborns under 37 weeks of gestational age compared to newborns who were 37 weeks or older. This is consistent with research from Gondar [15], Tikur Anbessa Specialized Hospital, Ethiopia [16] and Kenya [17], which found that preterm newborns had a higher mortality rate than term

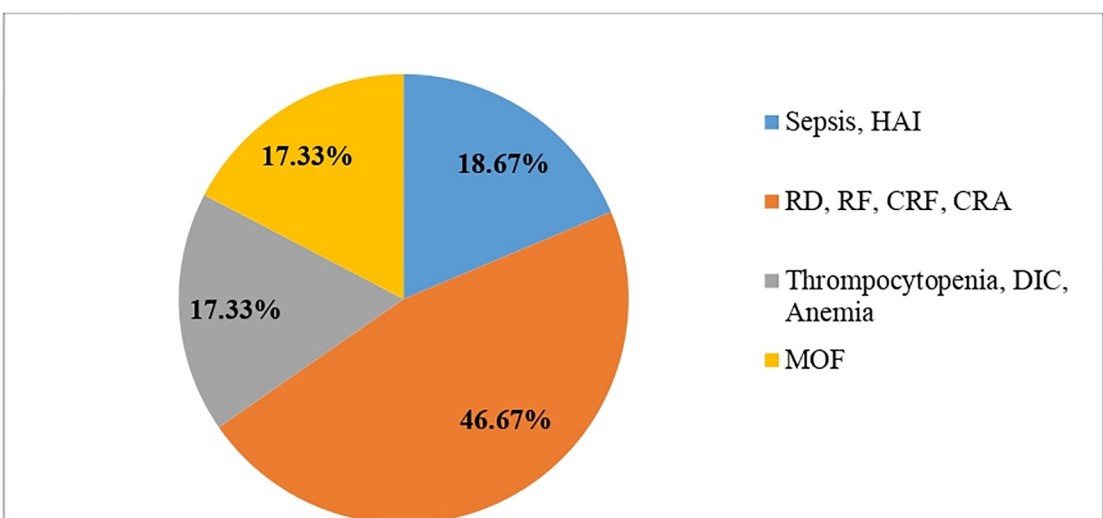

**Fig 2. The possible causes of neonatal deaths among neonates admitted to neonatal intensive care unit in public hospitals, Addis Ababa, Ethiopia, 2021.**

**Table 4. Bi-variable and multivariable logistic regression treatment outcome of neonatal sepsis among neonates admitted to neonatal intensive care unit in public hospitals, Addis Ababa, Ethiopia, 2021 (n = 308).**

| Variables | Neonatal sepsis | | COR (95% of CI) | AOR (95% OF CI) |
|---|---|---|---|---|
| | **Good** | **Poor** | | |
| Mothers Educ. Status | | | | |
| Can't read and write | 9(60.0%) | 6(40.0%) | 2.89(0.94–8.89) | 1.29(0.16–10.08) |
| Primary Educ. | 46(64.8%) | 25(35.2%) | **2.36(1.22–4.55)*** | 1.34(0.41–4.39) |
| Secondary Educ. | 74(78.7%) | 20(21.3%) | 1.17(0.60–2.28) | 1.86(0.59–5.79) |
| College and higher | 104(81.3%) | 24(18.8%) | **1** | 1 |
| Place of delivery | | | | |
| Home | 2(40.0%) | 3(60.0%) | 2.93(0.45–18.91) | 0.59(0.02–19.23) |
| Hospital | 190(76.9%) | 57(23.1%) | **0.52(0.29–0.97)*** | 0.36(0.11–1.17) |
| Health center | 41(73.2%) | 15(26.8%) | 1 | 1 |
| Age of the newborn at admission NICU | 28(60.9%) | 18(39.1%) | **2.31(1.19–4.48)*** | 1.29(0.37–4.45) |
| <30 minutes | 205(78.2%) | 57(21.8%) | 1 | 1 |
| ≥30 minutes | | | | |
| Gestational Age | | | | |
| <37 Wks | 102(64.6%) | 56(35.4%) | **3.53(1.99–6.27)*** | **4.87(1.23–19.22)**\*\* |
| ≥37 Wks | 131(87.3%) | 19(12.7%) | 1 | 1 |
| Birth weight group | | | | |
| <2.5kg | 98(81.6%) | 47(18.4%) | **2.31(1.35–3.95)*** | 0.33(0.08–1.30) |
| ≥2.5kg | 135(82.8%) | 28(17.2%) | 1 | 1 |
| Hypertension PIH/ Eclampsia | | | | |
| Yes | 163(68.2%) | 76(31.8%) | **1.96(1.06–3.64)*** | **3.54(1.24–10.09)**\*\* |
| No | 143(79.0%) | 38(21.0%) | 1 | 1 |
| PROM | | | | |
| Yes | 122(71.8%) | 48(28.2%) | **2.78(1.36–5.68)*** | **3.66(1.20–11.15)**\*\* |
| No | 81(90.0%) | 9(10.0%) | 1 | 1 |
| Meconium amniotic stained | | | | |
| Yes | 32(45.7%) | 38(54.3%) | **6.45(3.59–11.59)*** | **3.03(1.02–9.01)**\*\* |
| No | 201(84.5%) | 37(15.5%) | 1 | 1 |
| Chorioamnionitis | | | | |
| Yes | 36(85.7%) | 6(14.3%) | **0.55(0.22–1.37)*** | 2.50(0.57–10.99) |
| No | 167(76.6%) | 51(23.4%) | 1 | 1 |
| Sever chest in drawing | | | | |
| Yes | 123(69.9%) | 53(30.1%) | **2.83(1.57–5.10)*** | 1.85(0.56–6.09) |
| No | 110(83.3%) | 22(16.7%) | 1 | 1 |
| Grunting | | | | |
| Yes | 134(65.4%) | 71(34.6%) | **4.80(2.35–9.81)*** | **6.94(1.48–32.54)**\*\*\* |
| No | 99(96.1%) | 4(3.9%) | 1 | 1 |
| Un able to feed | | | | |
| Yes | 165(74.0%) | 58(26.0%) | **2.16(1.09–4.27)*** | 1.82(0.54–5.79) |
| No | 68(80.0%) | 17(20.0%) | 1 | **1** |
| Temperature | | | | |
| Yes | 141(82.5%) | 30(17.5%) | **0.44(0.26–0.74)*** | 0.47(0.19–1.17) |
| No | 92(67.2%) | 45(22.8%) | 1 | 1 |
| **CBC_ result** | | | | |
| Normal | 92(81.4%) | 21(18.6%) | **0.59(0.34–1.05)*** | 0.42(0.15–1.22) |
| Raise/drop | 141(72.3%) | 54 (27.7%) | 1 | 1 |

(*Continued*)

**Table 4.** (Continued)

| Variables | Neonatal sepsis | | COR (95% of CI) | AOR (95% OF CI) |
|---|---|---|---|---|
| | Good | Poor | | |
| **CRP_ result** | | | | |
| Reactive | 176(77.2%) | 52(22.8%) | **2.68(1.52–4.69)**\* | **5.87(1.53–22.56)**\*\*\* |
| Non-reactive | 51(72.9%) | 19(27.1%) | 1 | 1 |
| Metronidazole | | | | |
| Yes | 10(30.3%) | 23(69.7%) | **9.86(4.43–21.98)**\* | 3.27(0.86–12.49) |
| No | 223(81.1%) | 52(18.9%) | 1 | 1 |
| Meropenum | | | | |
| Yes | 18(42.9%) | 24(57.1%) | **5.62(2.84–11.3)**\* | **4.16(1.22–14.21)**\*\* |
| No | 215(80.8%) | 51(19.2%) | 1 | 1 |
| Vancomaycine | | | | |
| Yes | 24(53.3%) | 21(46.7%) | **3.39(1.75–6.54)**\* | 2.01(0.56–7.19) |
| No | 209(79.5%) | 54(20.5%) | 1 | 1 |

**Key** 1 = Reference

\* Statistically significant by COR at p-value ≤0.25

\*\*Statistically significant by AOR at p-value <0.05

\*\*\*strongly associated by AOR: p-value<0.01

newborns. This result is comparable to that of an Australian study, which discovered that a gestational age of one week enhanced the neonatal survival rate by more than 5% [18].

The odds of neonates born from mothers with developing Hypertensive PIH/ Eclampsia were 4 times higher than those of neonates born from who did not develop Hypertensive PIH/ Eclampsia. This finding is consistent with previous research that found chronic hypertension to be a risk factor for neonatal sepsis in Ethiopia. This could be because maternal hypertensive problems have a direct impact on fetal wellbeing in the uterus, which contributes to neonatal sepsis at birth.

PROM was statistically significantly associated with a poor treatment outcome of sepsis. Mothers who gave birth to neonates with PROM were 4 times more likely to suffer from sepsis compared with those neonates born from women who had not developed PROM. This finding is comparable with studies conducted in Nepal [19], Mexico [20] and USA [21]. These could be caused by aerobic and anaerobic pathogens colonizing the birth canal, resulting in ascending amniotic fluid infection and neonate colonization at birth. Mother-to-fetus transmission of bacterial agents infecting the amniotic fluid and birth canal during labor and delivery may occur more frequently, resulting in neonatal sepsis (EONS) [22].

Meconium amniotic stained were 3 times developed poor treatment outcome when compared to those neonates without history of Meconium amniotic stained. Which is similar with a study in Bahir dar [12], Uganda [23], Ghana [3], India [24] and Nepal [19]. This is revealed that after meconium aspiration strict follow up is needed for neonates. This may be due to neonates delivered from women with meconium stained amniotic fluid are more liable to aspirate it and fill smaller air ways and alveoli in the lung. And it increases the multiplication of microbes that cause sepsis and predisposes to late onset neonatal sepsis (LONS) [25].

A neonate who has at risk of respiratory problem was significantly associated with poor outcome of neonatal sepsis. Those neonates developed with grunting were 7 times developed poor outcome compared to neonates without respiratory distress syndrome. This is similar with a study in Bahir dar [12]. This result comparable with studies done in Uganda [23] and Sudan

[26]. This may be due to health workers' ignorance of the syndrome's poor early detection of signs, and another explanation may be due to mothers' delay coming in coming to health facilities or institutions.

C-reactive protein levels were found to be significantly associated with a poor sepsis outcome. Positive CRP laboratory results of neonates were 6 times they developed poor treatment outcomes when compared to those of neonates' negative CRP results. This finding is similar with studies done in Nepal [19]. This could be because CRP is the most sensitive and widely used test, but it is necessary to consider a sepsis panel of at least three tests, at least two of which must be positive for one to suspect septicemia with reasonable certainty.

In most developing countries, including Ethiopia, empirical treatment is the primary method of managing neonatal sepsis. When compared to neonates who did not receive Meropenum, those who received it had a fourfold worse treatment outcome. In this hospital, antimicrobial use is primarily empirical. This may promote the development of resistant bacteria, influencing future drug selection in the treatment of neonatal sepsis.

## Conclusion and recommendation

The study found both maternal and neonatal factors as possible independent risk factors to have a strong association with the risk of poor outcome of neonatal sepsis. Preterm babies admitted to NICU, grunting, meconium amniotic stained (MSAF), duration of rupture of membrane >18hours (PROM), Hypertensive PIH/ Eclampsia, meropenum and CRP result were significantly associated with poor treatment outcome of neonatal sepsis. Researchers who are interested in conducting research on neonatal sepsis should have to include neonates in the community, which may increase the external validity of the study. It is also better to do a meta-analysis since the previous findings about the factors causing neonatal sepsis were inconsistent.

## Acknowledgments

We would like to Acknowledge Addis Ababa University, college of health sciences, school of nursing and Midwifery and department of nursing for giving me the chance. In addition, we would like to extend our thanks to the study participants, data collectors, supervisors for their contribution and commitment throughout the study period.

### Ethics approval and consent to participate

Ethical clearance was obtained from the institutional review board of Addis Ababa University, College of Health Sciences, School of Nursing and Midwifery, Department of Nursing (Protocol number: aau/chs/chnsg/18/21). Formal letters were obtained from Addis Ababa Public Health Research and Emergency Management Core Process in order to get permission to carry out the study. After explaining the purpose and procedure of the study, each respondent (mothers/care givers) signed a written informed consent form. No name or other identifying information was included with the instrument. The eligible study participants were enrolled in the study only after they gave written informed consent and will not be forced to participate. All the information given by the respondents was used for research purposes only; Confidentiality and privacy were maintained by omitting the names of the respondents during the data collection procedure and after the data collection was completed.

## Author Contributions

**Conceptualization:** Esmelealem Mihretu.

**Data curation:** Asalef Endazanaw, Yohannes Godie, Dires Birhanu.

**Formal analysis:** Yohannes Godie, Dires Birhanu.

**Investigation:** Asalef Endazanaw.

**Methodology:** Fikertemariam Abebe, Yohannes Godie, Yitayal Guadie, Esmelealem Mihretu.

**Software:** Tefera Mulugeta, Yitayal Guadie, Dires Birhanu.

**Supervision:** Tefera Mulugeta.

**Validation:** Yohannes Godie, Yitayal Guadie.

**Visualization:** Fikertemariam Abebe.

**Writing – original draft:** Asalef Endazanaw.

**Writing – review & editing:** Fikertemariam Abebe, Dires Birhanu, Esmelealem Mihretu.

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
