## [Decision Letter · Decision Letter 0]

31 Oct 2022

PONE-D-22-21579Treatment outcome of neonatal sepsis and associated factors among neonates admitted to neonatal intensive care unit in public hospitals of Addis Ababa City, Ethiopia, 2021: Cross-sectional studyPLOS ONE

Dear Dr. Mihretu,

Thank you for submitting your manuscript to PLOS ONE. After careful consideration, we feel that it has merit but does not fully meet PLOS ONE’s publication criteria as it currently stands. Therefore, we invite you to submit a revised version of the manuscript that addresses the points raised during the review process.

The reviewers have provided extensive comments to improve your manuscript.In addition, I have the following feedback. 

1.It is not evident from the methods section on how exactly the sample was systematically randomly sampled. Was the sampling based on the sequence of admissions into the NICU (Clarify)? Was it restricted to in house deliveries?

Based on your objectives, it appears the qualifying criteria to be included was 'Neonatal sepsis'. I would assume the gestational age at appearance of symptoms would be a very important predictor beyond just 30 minutes or more. 2. The analysis lacks a causal framework. Hence this results in the addition of too many and unnecessary variables in the model. Please draw a DAG to visualise what variables need to be in the model. The present model suggests that antibiotics cause a higher odds of poor outcome - antibiotics administration is correlated with the severity of the condition and should not probably be in the model.

3. Also, the concept of time to outcome is an important element in the model due to varying reasons. Hence a Cox proportional model or a similar time to event analytical model should have been used.

We look forward to receiving your revised manuscript.

Kind regards,

Diwakar Mohan, MD

Academic Editor

PLOS ONE

Journal Requirements:

2.  In the ethics statement in the Methods, you have specified that verbal consent was obtained. Please provide additional details regarding how this consent was documented and witnessed, and state whether this was approved by the IRB.

3. You indicated that you had ethical approval for your study. Please clarify whether minors (neonate-mothers below the age of 18 years) were included in the study. If yes, in your Methods section, please ensure you have also stated whether you obtained consent from parents or guardians of the minors included in the study or whether the research ethics committee or IRB specifically waived the need for their consent.

4. We suggest you thoroughly copyedit your manuscript for language usage, spelling, and grammar. If you do not know anyone who can help you do this, you may wish to consider employing a professional scientific editing service.

The name of the colleague or the details of the professional service that edited your manuscriptA copy of your manuscript showing your changes by either highlighting them or using track changes (uploaded as a *supporting information* file)A clean copy of the edited manuscript (uploaded as the new *manuscript* file)”

5. PLOS requires an ORCID iD for the corresponding author in Editorial Manager on papers submitted after December 6th, 2016. Please ensure that you have an ORCID iD and that it is validated in Editorial Manager. To do this, go to ‘Update my Information’ (in the upper left-hand corner of the main menu), and click on the Fetch/Validate link next to the ORCID field. This will take you to the ORCID site and allow you to create a new iD or authenticate a pre-existing iD in Editorial Manager. Please see the following video for instructions on linking an ORCID iD to your Editorial Manager account: https://www.youtube.com/watch?v=_xcclfuvtxQ.

6. We note that you have indicated that data from this study are available upon request. PLOS only allows data to be available upon request if there are legal or ethical restrictions on sharing data publicly. For more information on unacceptable data access restrictions, please see http://journals.plos.org/plosone/s/data-availability#loc-unacceptable-data-access-restrictions.

7. Please upload a new copy of Figures 1 and 2 as the detail is not clear. Please follow the link for more information: " ext-link-type="uri" xlink:type="simple">https://blogs.plos.org/plos/2019/06/looking-good-tips-for-creating-your-plos-figures-graphics/"
https://blogs.plos.org/plos/2019/06/looking-good-tips-for-creating-your-plos-figures-graphics/

Reviewers' comments:

Reviewer's Responses to Questions

**Comments to the Author**

1. Is the manuscript technically sound, and do the data support the conclusions?

Reviewer #1: Partly

Reviewer #2: Partly

2. Has the statistical analysis been performed appropriately and rigorously? 

Reviewer #1: Yes

Reviewer #2: Yes

3. Have the authors made all data underlying the findings in their manuscript fully available?

Reviewer #1: No

Reviewer #2: Yes

4. Is the manuscript presented in an intelligible fashion and written in standard English?

Reviewer #1: Yes

Reviewer #2: No

5. Review Comments to the Author

Reviewer #1: First of all, I admit this is an important manuscript focusing on an important issue and question in neonatal health research. There are several major aspects where authors need to clarify (and/or need to make revisions) in the manuscript, as follows:

1. in Line 24: it says - "308 admitted neonates were selected". Text in line 172 it also says: "A total of 308 neonatal-mothers interviewed" whereas in Line 105 - the text shows: "There were 724 eligible admitted neonates, and respondents were interviewed on discharge using a pretested interviewer-administered questionnaire".

Please clarify which is the correct number of respondents interviewed; why instead of all 724 eligible admitted cases only 308 were interviewed? Who were excluded and why? What is the potential impact of such exclusion on the result and how do you explain the impact for limiting your analysis to your N=308 (instead of 724)

2. Text in line 78 said the study was carried out "among neonates admitted to NICU wards in Addis Ababa city public hospitals"; line 82 says: "The city has twelve public Hospitals"; and then in line 100 text says "Of the Eleven public hospitals in Addis Ababa city". Please ensure consistency and mention the correct number of public hospitals in Addis Ababa city.

3. From text in Line 78 and 82 - it apparently appears as the study was conducted in all public hospitals in Addis Ababa city - but the text in line 90-91 clarifies as " the study was conducted in four Addis Ababa public hospitals selected by lottery method" (similar text also appears in line 100-101 : "Of the Eleven public hospitals in Addis Ababa city, four public hospitals (36%) were randomly selected for this study using a lottery method."

The text in the abstract, and in the lines of 78 and 82 gave a wrong impression to reader that the study was conducted in all city public hospital - which is actually conducted in selected 4 city public hospitals (as confirmed in line 90-91 and 100-101). This needs to be revised and clear to readers.

4. Text in line 101-103 says: "The total number of neonates diagnosed as sepsis and admitted to NICU per month in each of the selected hospitals was reviewed prior to the actual data collection." Question is: For how long did you review the "per month" number of admission? Whether you reviewed this monthly admission number for last six month? or for last one year? or did you review monthly admission of same months in last year (mid-February - mid-May in 2020)? That's may be an important aspect - considering the seasonal variation of sickness and admission of sick neonate in the hospital.

5. Text in line 26 describes the analysis method as "binary logistic regression" and line 153-156 says about multivariate logistic regression method. Please ensure consistency and revise the text as needed.

6. Text in line 166 mentioned as "Oral informed consent was obtained". Is there strong rationale for not taking written consent from the respondents? if yes - what are those?

7. The analysis shows Preterm (gestational age less than 37 weeks) and Low birth weight (LBW) as separate variables and authors fit these two in the analysis model which revealed significant impact of preterm on treatment outcome while non-significant impact on outcome. My question is - did you check impact for combined preterm and LBW on treatment outcome? That analysis could reveal important result and may lead to suggest for health program manager/planner.

8. One weakness in the discussion part of the manuscript is lacking focus on strengths and limitations of the study. I suggest the authors to include a paragraph describing their thoughts about the strength and limitations of the study they conducted and presented.

9. Couple of additional questions/comments, I like to offer to the authors:

- a) Have you collected the time/duration between the time when symptoms appeared (i.e., the time when the neonate became sick) and the time when the sick neonate got admitted in the NICU? A sick neonate admitted earlier stage vs a sick neonate admitted at terminal stage might have huge impact on the treatment outcome - you must not ignore that

- b) have you scored the severity of disease/sepsis status among the neonates you enrolled in the analysis?

Both severity of sepsis and timing of admission would have plausible impact on treatment outcome - therefore, controlling for these two variables into the regression model is critically important before the authors suggest recommendations to readers.

Reviewer #2: Comments to authors

1. On the title and abstract part:

The title needs modification and be more specific because it just measured the factors contributing to the poor treatment due to neonatal sepsis, it also should be specific outcome either mortality, complications, worsened and so on among the poor outcomes.

In abstract, the main gap in knowledge is missing; the aims and the rationale of the study have not been explicitly mentioned.

The method section is not clear and adequate. In conclusion, the sentence "Gestational age, grunting, meconium amniotic stained, hypertensive, meropenum and C-reactive protein level were significantly associated with treatment outcome of neonatal sepsis" is not clear, which treatment outcome? Which is the most important finding of this study? The recommendation of the results of the study is missed.

2. In the introduction part:

The way the introduction has been written is not good, it is better to include risk factors contributing to poor treatment outcome. It talks about specifically mortality not poor outcome Also, the main gap in the knowledge and the rationales of the study are missed.

3. Material and Methods:

The method part is not clear and lacking many things such as; Inclusion criteria are not mentioned, algorithm for sample allocation, sepsis diagnosis and treatment guidelines in NICU among the four specific hospital.

4. Result part:

The results of this study lacks many things are missed such as clinical parameters, types of sepsis, neonatal related characteristics, laboratory findings, comorbidity, and medication related factors,

Etiology of sepsis is entirely missed.

The result among four hospitals are not separately described

5. Discussion

The discussion section is poor, the possible justification is also poor.

It should be re-written in more coherent and concise way with logical explanation

6. Conclusion and recommendation

The conclusion is not written strongly. The recommendation is totally missed.

7. Figures and tables

Table-2 should be divided in to two tables; some findings should be described with figures

8. Does the paper raise any concerns?

There are no major concerns raised by the paper.

There seems no ethical concern.

Under statistical analysis, why did you do logistic regression if your primary outcome is mortality among poor treatment outcome, rather cox regression is preferred.

References need some modifications that for some reference there is ‘et al’ where as none for some reference.

9. Competing interest

There is no competing interest

10. English editing

The manuscript requires English editing to correct the grammar or flow.

Recommendations to the Editor:

Better to modify the manuscript as per the comments above.

No additional comment

6. PLOS authors have the option to publish the peer review history of their article (what does this mean?). If published, this will include your full peer review and any attached files.

Reviewer #1: **Yes: **Rashed Shah

Reviewer #2: No

---

## [Author Response · Author response to Decision Letter 0]

3 Feb 2023

Point by point response 

Treatment outcome of neonatal sepsis and associated factors among neonates admitted to neonatal intensive care unit in public hospitals of Addis Ababa City, Ethiopia, 2021

Reviewers' comments: 

Reviewer's Responses to Questions

1. Is the manuscript technically sound, and do the data support the conclusions?

Reviewer #1: Partly

Reviewer #2: Partly

Response: We act accordingly with thanks 

2. Has the statistical analysis been performed appropriately and rigorously?

Reviewer #1: Yes

Reviewer #2: Yes

Response: accepted with thanks

3. Have the authors made all data underlying the findings in their manuscript fully available?

Reviewer #1: No

Reviewer #2: Yes

Response : Thanks, and we made it fully available 

4. Is the manuscript presented in an intelligible fashion and written in standard English?

Reviewer #1: Yes

Reviewer #2: No

Response : We presented accordingly 

5. Review Comments to the Author

Comment: Reviewer #1: First of all, I admit this is an important manuscript focusing on an important issue and question in neonatal health research. There are several major aspects where authors need to clarify (and/or need to make revisions) in the manuscript, as follows:

1. In Line 24: it says - "308 admitted neonates were selected". Text in line 172 it also says: "A total of 308 neonatal-mothers interviewed" whereas in Line 105 - the text shows: "There were 724 eligible admitted neonates, and respondents were interviewed on discharge using a pretested interviewer-administered questionnaire".

Please clarify which is the correct number of respondents interviewed; why instead of all 724 eligible admitted cases only 308 were interviewed? Who were excluded and why? What is the potential impact of such exclusion on the result and how do you explain the impact for limiting your analysis to your N=308 (instead of 724)

Response: We appreciate this idea of insight. The number ‘724’ was a total number of neonates diagnosed as sepsis and admitted to NICU per month in each of the selected hospitals. Among thus, we select a total of 308 neonates using systematic random sampling technique and our analysis was depending on these sample size (308). 

2. Text in line 78 said the study was carried out "among neonates admitted to NICU wards in Addis Ababa city public hospitals"; line 82 says: "The city has twelve public Hospitals"; and then in line 100 text says "Of the Eleven public hospitals in Addis Ababa city". Please ensure consistency and mention the correct number of public hospitals in Addis Ababa city.

Response: Really, there are 12 public hospitals in Addis Ababa city in general. Among thus, one hospital (Amanuel hospital) hasn’t NICU service. Only 11 public hospitals are candidate for our studies. Therefore, four public hospitals were selected randomly among eleven public hospitals that have NICU services in the Addis Ababa city. 

3. From text in Line 78 and 82 - it apparently appears as the study was conducted in all public hospitals in Addis Ababa city - but the text in line 90-91 clarifies as " the study was conducted in four Addis Ababa public hospitals selected by lottery method" (similar text also appears in line 100-101: "Of the Eleven public hospitals in Addis Ababa city, four public hospitals (36%) were randomly selected for this study using a lottery method."

The text in the abstract, and in the lines of 78 and 82 gave a wrong impression to reader that the study was conducted in all city public hospital - which is actually conducted in selected 4 city public hospitals (as confirmed in line 90-91 and 100-101). This needs to be revised and clear to readers.

Response: Thank you, we made it clear for readers. In general, there are 12 public hospitals in Addis Ababa city and eleven hospitals have NICU services. So our study was focuses on those public hospitals that have NICU services (i.e., only eleven hospitals.)

4. Text in line 101-103 says: "The total number of neonates diagnosed as sepsis and admitted to NICU per month in each of the selected hospitals was reviewed prior to the actual data collection." Question is: For how long did you review the "per month" number of admission? Whether you reviewed this monthly admission number for last six month? or for last one year? or did you review monthly admission of same months in last year (mid-February - mid-May in 2020)? That's may be an important aspect - considering the seasonal variation of sickness and admission of sick neonate in the hospital.

Response: We appreciate this idea of insight. "The total number of neonates diagnosed as sepsis and admitted to NICU per month in each of the selected hospitals was reviewed for one month of the same months in last year i.e., from February 15 to May 10, in 2020. It is recommended to take the same durations of the previous year to minimize the seasonal variation of sickness and admission of sick neonate in the hospitals.

5. Text in line 26 describes the analysis method as "binary logistic regression" and line 153-156 says about multivariate logistic regression method. Please ensure consistency and revise the text as needed.

Response: Thank you; we use binary logistic regression to identify the candidate variables for multivariate logistic regressions and those variables which have “p 0.25” in bivariate logistic regression were again used in multivariate logistic regressions to identify significant variables for treatment outcomes. 

6. Text in line 166 mentioned as "Oral informed consent was obtained". Is there strong rationale for not taking written consent from the respondents? If yes - what are those?

Response: Sorry, that was typing error we obtain written consent from each participants. See the revised manuscript

7. The analysis shows Preterm (gestational age less than 37 weeks) and Low birth weight (LBW) as separate variables and authors fit these two in the analysis model which revealed significant impact of preterm on treatment outcome while non-significant impact on outcome. My question is - did you check impact for combined preterm and LBW on treatment outcome? That analysis could reveal important result and may lead to suggest for health program manager/planner.

Response: We appreciate this idea of insight. As WHO define preterm is the birth of a baby that occurs before 37 weeks of gestation regardless of their weight, we haven’t seen combined effect on treatment outcome; we have checked cofounding variables between preterm and LBW via tolerance test. Actually to know the impact of combined preterm and LBW on treatment outcomes we have to use prediction model which is not our aim. However, this is an interesting issue and we will see in the other future studies.

8. One weakness in the discussion part of the manuscript is lacking focus on strengths and limitations of the study. I suggest the authors to include a paragraph describing their thoughts about the strength and limitations of the study they conducted and presented.

Response: Thank you; we made it.

9. Couple of additional questions/comments, I like to offer to the authors:

- a) Have you collected the time/duration between the time when symptoms appeared (i.e., the time when the neonate became sick) and the time when the sick neonate got admitted in the NICU? A sick neonate admitted earlier stage vs a sick neonate admitted at terminal stage might have huge impact on the treatment outcome - you must not ignore that

- b) have you scored the severity of disease/sepsis status among the neonates you enrolled in the analysis?

Both severity of sepsis and timing of admission would have plausible impact on treatment outcome - therefore, controlling for these two variables into the regression model is critically important before the authors suggest recommendations to readers.

Response: We adorations this idea of insight. We haven’t seen because this study is not a follow up study. Simply we observe the final out comes of the neonate admitted with sepsis and what were the significant factors for that outcome. 

Reviewer #2: Comments to authors

1. on the title and abstract part:

The title needs modification and be more specific because it just measured the factors contributing to the poor treatment due to neonatal sepsis, it also should be specific outcome either mortality, complications, worsened and so on among the poor outcomes.

In abstract, the main gap in knowledge is missing; the aims and the rationale of the study have not been explicitly mentioned.

The method section is not clear and adequate. In conclusion, the sentence "Gestational age, grunting, meconium amniotic stained, hypertensive, meropenum and C-reactive protein level were significantly associated with treatment outcome of neonatal sepsis" is not clear, which treatment outcome? Which is the most important finding of this study? The recommendation of the results of the study is missed.

Response: We respect this idea of insight. We made a correction and you can see in the revised manuscript. Regarding to the title we focused on the general outcome of neonatal sepsis and which variables had significant factors for the occurrence of that outcomes. The outcomes were neonates discharge (with treatment success), leave against medical advice (LAMA), referred and death. We tried to describe that how many percent of each treatment outcome were observed in this study. 

2. in the introduction part:

The way the introduction has been written is not good, it is better to include risk factors contributing to poor treatment outcome. It talks about specifically mortality not poor outcome Also, the main gap in the knowledge and the rationales of the study are missed.

Response: Accepted with thank you and see on the revised manuscript 

3. Material and Methods:

The method part is not clear and lacking many things such as; Inclusion criteria are not mentioned, algorithm for sample allocation, sepsis diagnosis and treatment guidelines in NICU among the four specific hospital.

Response: Accepted with thank you and see on the revised manuscript

4. Result part:

The results of this study lacks many things are missed such as clinical parameters, types of sepsis, neonatal related characteristics, laboratory findings, comorbidity, and medication related factors,

Etiology of sepsis is entirely missed.

The result among four hospitals are not separately described

5. Discussion

The discussion section is poor, the possible justification is also poor.

It should be re-written in more coherent and concise way with logical explanation

Response: Accepted with thank you and see on the revised manuscript

6. Conclusion and recommendation

The conclusion is not written strongly. The recommendation is totally missed.

Response: Thank you; we made it accordingly and see on the revised manuscript

7. Figures and tables

Table-2 should be divided in to two tables; some findings should be described with figures

8. Does the paper raise any concerns?

There are no major concerns raised by the paper.

There seems no ethical concern.

Under statistical analysis, why did you do logistic regression if your primary outcome is mortality among poor treatment outcome, rather cox regression is preferred.

References need some modifications that for some reference there is ‘et al’ where as none for some reference.

9. Competing interest

There is no competing interest

10. English editing

The manuscript requires English editing to correct the grammar or flow.

Recommendations to the Editor:

Better to modify the manuscript as per the comments above.

No additional comment

---

## [Editor Report · Decision Letter 1]

13 Apr 2023

Treatment outcome of neonatal sepsis and associated factors among neonates admitted to neonatal intensive care unit in public hospitals, Addis Ababa, Ethiopia, 2021. Multi-center cross-sectional study

PONE-D-22-21579R1

Dear Dr.Esmelealem Mihretu ,

We’re pleased to inform you that your manuscript has been judged scientifically suitable for publication and will be formally accepted for publication once it meets all outstanding technical requirements.

Kind regards,

Sanjoy Kumer Dey, M.D

Academic Editor

PLOS ONE
---

## [Editor Report · Acceptance letter]

18 May 2023

PONE-D-22-21579R1 

Treatment outcome of neonatal sepsis and associated factors among neonates admitted to neonatal intensive care unit in public hospitals, Addis Ababa, Ethiopia, 2021. Multi-center cross-sectional study 

Dear Dr. Mihretu:

I'm pleased to inform you that your manuscript has been deemed suitable for publication in PLOS ONE. Congratulations! Your manuscript is now with our production department. 

Kind regards, 

on behalf of

Dr. Sanjoy Kumer Dey 

Academic Editor

PLOS ONE